# Metaverse Architectures: Hypernetwork and Blockchain Synergy

**Janne Ruponen**[1], **Ivan Dorokhov**[1], **Sergey Barykin**[2], **Sergey Sergeev**[2] and **Andrey Nechesov**[1]. *

[1] Artificial Intelligence Research Center of Novosibirsk State University, Novosibirsk, Russia
[2] Peter the Great St.Petersburg Polytechnic University

ruponez@gmail.com ioandorokhov@gmail.com; sbe@list.ru;
sergeev2@yandex.ru nechesoff@gmail.com;

## Abstract

We propose a Hypernetwork-driven digital twin framework for the metaverse, integrating AI-driven predictions with time-exact blockchain synchronization to enhance urban modeling accuracy and virtual ecosystem performance. Our system employs a hierarchical blockchain architecture with temporal verification mechanisms to ensure secure and scalable data exchange in real time. By embedding polynomial-time verification into IoT data pipelines and simulation engines, we tackle critical challenges in resource interaction, security, and privacy within metaverse-based infrastructures. This approach establishes a trustworthy, adaptive, and decentralized ecosystem for digital twins, advancing next-generation smart city innovations.

## 1 Introduction

Modern urban environments are increasingly driven by digital transformation, where physical and virtual processes Nechesov et al. (2025) must be managed seamlessly. Emerging paradigms like the metaverse Iqbal & Campbell (2023); Lee et al. (2021) promise to integrate these processes, but doing so effectively requires novel frameworks that unify disparate data flows and management systems.

### 1.1 Overview of Our Framework

To provide a clear, step-by-step understanding of our approach, we summarize our framework as follows:

1. **Resource Flow Integration via Hypernetworks:** Our system models material, financial, and informational flows as interconnected layers within a hypernetwork. This enables dynamic, real-time interactions and lays the foundation for a spatial economy.

2. **Hierarchical Multi-Blockchain Architecture:** We deploy specialized blockchain Goncharov & Nechesov (2023b) ecosystems for each resource type, ensuring secure, time-exact synchronization. The use of polynomial-time Nechesov & Goncharov (2024) verification (i.e., $O(n^k)$ for a fixed constant $k$) guarantees scalability even as data volumes increase.

3. **Adaptive Data Fidelity and Dynamic Assembly:** Leveraging reinforcement learning, our system dynamically adjusts the fidelity of data sharing. It assembles temporary blockchains via context-aware stakeholder routing to process high-fidelity data during critical events.

4. **Cognitive and Semantic Coordination:** The decentralized cognitive layer (cognitome) and the semantic layer (logisticon) work in tandem to predict, regulate, and coordinate resource flows. This integration enables robust digital twin modeling of urban environments.

*This work was supported by a grant for research centers, provided by the Analytical Center for the Government of the Russian Federation in accordance with the subsidy agreement (agreement identifier 000000D730324P540002) and an agreement with the Novosibirsk State University dated 27 December 2023 No. 70-2023-001318.

5. **Hierarchical Governance and Scalability:** Our framework scales seamlessly from individual agents through corporate entities and cities to global systems, supporting decentralized governance across multiple levels.

This layered approach ensures real-time data synchronization, predictive accuracy, and adaptive control—paving the way for next-generation smart city innovations.

## 1.2 PAPER ORGANIZATION

The remainder of the paper is organized as follows. Section 2 reviews key concepts including the metaverse, hypernetworks, and the roles of the cognitome and logisticon. Section 3 introduces our hypernetwork framework for integrating resource flows, while Section 4 details the dynamic multiblockchain formation. Section 5 describes the construction of the Logisticon Hypernetwork via decentralized cognitive and semantic coordination. In Section 6, we extend the framework to hierarchical systems for smart city ecosystems. Section 7 presents our AI-driven digital twin framework and real-time data integration, and Section 8 discusses the system design and implementation aspects. Section 9 provides the mathematical formulation and algorithmic synthesis of the entire framework, and finally, Section 10 concludes the paper with a discussion of its implications and future directions.

## 2 BACKGROUND AND KEY CONCEPTS

To understand our approach, it is essential to grasp several key concepts that underpin our framework. We propose the definitions of Metaverse, Hypernetworks, Cognitome and a new concept of Logisticon (note that some definitions may differ from conventional usage). In this section, we introduce these concepts in a narrative form, providing both context and definitions.

**Metaverse:** The metaverse is a hybrid spatial-economic environment that integrates physical and virtual processes. It achieves this by orchestrating flows of information, material resources, and financial transactions through dynamic scenarios and adaptive algorithms, thereby merging the real and digital worlds into a single managed system. Kitchin's analysis of the data revolution underscores how the emergence of big data and open data infrastructures can serve as the backbone for the metaverse, enabling interconnected digital ecosystems that facilitate real-time, data-driven interactions and innovative value creation. Kitchin (2014).

**Hypernetworks:** Hypernetworks are multilayer structures that facilitate the simultaneous interaction of diverse resource flows. They employ intelligent coordination mechanisms to dynamically adapt and synchronize material, financial, and informational streams. Our framework leverages hypernetworks to build a cohesive infrastructure that supports both physical processes and virtual models Barykin et al. (2024). Regarding the holistic approach to the consideration of different material, financial, and informational flows, Ha et al. demonstrate how hypernetworks can dynamically generate and adapt network parameters within multilayer architectures, a concept that parallels our use of hypernetworks to intelligently coordinate and synchronize diverse resource flows—material, financial, and informational—into a unified infrastructure supporting both physical processes and virtual models.Ha et al. (2016) In relation to the research of hypernetworks, Battiston et al. provide a comprehensive framework for understanding interactions that extend beyond simple pairwise connections, thereby laying the theoretical groundwork for hypernetworks as complex, multi-dimensional structures that capture higher-order interdependencies in diverse systems Battiston (2020)

**Cognitome:** Based on P.K. Anokhin's work Anokhin (1975), and the consideration of Anokhin's approach by Lapkin et al. (2018) the cognitome is a functional system for processing and storing information that provides predictive management of processes. In our context, the cognitome unifies the three levels of resource flows—material, financial, and informational—into a single cognitive management system. This system not only analyzes but also forecasts process developments, enabling proactive decision-making.

**Logisticon:** The logisticon is a new concept of the semantic system designed to describe logistic processes by converting various aspects of resource flows into structured language models. It serves as a bridge between logistic operations and cognitive technologies, effectively acting as input for large language models (LLMs) and AI systems. In our framework, the logisticon is central to

establishing clear and actionable rules for resource coordination. Gruber's translation approach to portable ontology specifications lays a conceptual foundation for the logisticon, a semantic system that translates diverse aspects of resource flows into structured language models, thereby bridging logistic operations and cognitive technologies for effective resource coordination. Gruber (1993).

**Logistics and Service Metaverse:** Our concept of the logistics and service metaverse extends beyond traditional VR/AR environments. It is a unified spatial-economic framework that incorporates:

- *Physical Processes:* such as the movement of goods and the operation of real-world infrastructure.
- *Virtual Process Models:* including digital twins Goncharov & Nechesov (2023a), simulations, and predictive algorithms.
- *Hypernetwork Interaction:* which interlinks the above processes, ensuring that the physical and virtual realms continuously adapt to each other.

## 3  HYPERNETWORK: INTEGRATING RESOURCE FLOWS

### 3.1  RELEVANCE OF THE STUDY

Modern logistics and service systems are challenged by rapid digital evolution, requiring the integration of physical operations with advanced digital modeling. Our framework uses hypernetworks to connect material, financial, and informational flows in real time, forming the backbone of a spatial economy that optimizes resource utilization.

Time-exact multi-blockchain systems further enhance this integration by managing distinct data types (material records, financial transactions, and informational logs) via specialized chainsChristidis & Devetsikiotis (2016). This decentralized approach, combined with AI-driven dynamic stakeholder routing Nechesov & Ruponen (2024), ensures that our system is both efficient and scalable.

### 3.2  SCIENTIFIC PROBLEM

Traditional economic research often treats material, financial, and informational flows in isolation, which impedes the development of comprehensive management models. Our work addresses this gap by proposing a unified framework that merges these flows, enabling advanced AI integration for predictive and adaptive system management.

### 3.3  METAVERSE AS AN INTEGRATION OF FLOWS

Within our proposed framework, the metaverse serves as an integrated system where:

- **Information flows** (including data, digital twins, and optimization algorithms) overlay material and financial flows to generate dynamic interactions.
- The **spatial economy** is modeled through the co-evolution of industries and technological convergence.
- Both human and AI agents interact within the hypernetwork to drive decentralized decision-making and industrial co-evolution.

## 4  DYNAMIC MULTIBLOCKCHAIN HYPERNETWORK FORMATION

### 4.1  MULTI-BLOCKCHAIN ECOSYSTEMS

The hypernetwork is constructed from specialized blockchain ecosystems for each resource type:

- **Material Flows:** Represented by blockchains $\{B_{m,j}(t)\}_{j \in J_m}$.
- **Financial Flows:** Represented by blockchains $\{B_{f,k}(t)\}_{k \in J_f}$.
- **Information Flows:** Represented by blockchains $\{B_{i,l}(t)\}_{l \in J_i}$.

Let's define:

- $J_m$: The finite index set for blockchains handling material flows
- $J_f$: The finite index set for blockchains handling financial flows
- $J_i$: The finite index set for blockchains handling information flows

Each blockchain $B_{x,\alpha}(t)$ (with $x \in \{m, f, i\}$ and index $\alpha \in J_x$) evolves as:

$$h_{x,\alpha}^{(t+1)} = H\Big(h_{x,\alpha}^{(t)}, \phi_{x,\alpha}(t), \Psi_x(t)\Big),$$

where:

- $h_{x,\alpha}^{(t)}$ is the current state.
- $\phi_{x,\alpha}(t)$ represents the new data (e.g., transactions or sensor updates) input to blockchain $B_{x,\alpha}$ at time $t$.
- $\Psi_x(t)$ aggregates external influences or inputs from blockchains in other ecosystems.

$$\Psi_x(t) = \bigoplus_{y \neq x} w_{x,y} \Phi_y(t)$$

where

- $\bigoplus$ is a weighted aggregator
- $w_{x,y}$ is a weighting factor capturing the influence of ecosystem $y$ on ecosystem $x$.
- $\Phi_y(t)$ is an aggregation function over the new data inputs from all blockchains in ecosystem y:

$$\Phi_y(t) = \sum_{\alpha \in J_y} \phi_{y,\alpha}(t)$$

### 4.2 HYPERNETWORK INTEGRATION

The integrated ecosystem for each resource type is modeled as:

$$G_x(t) = \Big( \bigcup_{\alpha \in J_x} V_{x,\alpha}(t), \ \bigcup_{\alpha \in J_x} E_{x,\alpha}(t) \Big).$$

The overall hypernetwork is represented by the hypergraph:

$$H(t) = \Big( V(t), E(t) \Big),$$

with

$$V(t) = V_m(t) \cup V_f(t) \cup V_i(t),$$

where

$$V_x(t) = \bigcup_{\alpha \in J_x} V_{x,\alpha}(t)$$

and hyperedges defined as:

$$E(t) = \{e \subset V(t) : e \cap V_m(t) \neq \varnothing, \ e \cap V_f(t) \neq \varnothing, \ e \cap V_i(t) \neq \varnothing\}.$$

## 5 CONSTRUCTING THE LOGISTICON HYPERNETWORK: DECENTRALIZED COGNITOME AND DYNAMIC BLOCKCHAIN ASSEMBLY

### 5.1 ADAPTIVE DATA FIDELITY AND DYNAMIC BLOCKCHAIN ASSEMBLY

Our system employs a reinforcement learning (RL) algorithm to manage data fidelity and dynamically assemble blockchain groups:

1. **Adaptive Data Fidelity:** The RL agent monitors network load, event criticality, and resource availability to select between:
   - *Low-Fidelity Sharing:* Lightweight blockchains for routine textual updates.
   - *High-Fidelity Sharing:* High-fidelity blockchains for rich data streams during critical events.

2. **Dynamic Stakeholder Routing:** For high-fidelity sharing, a context-aware routing function is used:
$$\mathcal{R}(E, p, r) = \{j \in A : d(p_j, p_E) \leq \delta \ \wedge \ R_j \geq r_{\min}\},$$
   where
   - $E$ represent the set of all potential destinations in the hypernetwork
   - $p$ denotes the set of positions of the destinations in the hypernetwork
   - $r$ is a vector containing the routing requirements or constrains for each destination
   - $A$ is the set of all available paths in the the hypernetwork
   - $p_j$ denotes the position of destination $j$ within the set $E$.
   - $p_E$ is the position of the destination withing the set $E$.
   - $d(p_j, P_E)$ is the distance function that calculates the distance between the position $p_j$ and $p_E$
   - $\delta$ is a threshold distance. The path is considered valid if the distance beetween $p_j$ and $p_E$ is less than or equal to $\delta$.
   - $R_j$ is explicitly defined as the reliability score for agent $j$, computed from historical performance and availability metrics. This subset forms a temporary blockchain $B_{\text{temp}}$ for time-exact processing.

The RL update rule is:

$$Q(s_i(t), a) \leftarrow Q(s_i(t), a) + \eta \Big[ r(s_i(t), a) + \gamma \max_{a'} Q(s_i(t+1), a') - Q(s_i(t), a) \Big],$$

and the communication cost is modeled as:

$$C_i(t) = f_i(t)\, C_{\text{high}} + \big(1 - f_i(t)\big)\, C_{\text{low}},$$

with $f_i(t) = 1$ indicating high-fidelity mode.

## 5.2 DECENTRALIZED COGNITOME AND LOGISTICON FRAMEWORK

The cognitome is the decentralized cognitive layer that coordinates material, financial, and informational flows and provides predictive control over processes. The Cognitome functions as an autonomous decision-making network that:

1. Predictive Flow Management
   - Uses machine learning to optimize logistics in real-time
   - Continuously adjusts material deliveries, financial settlements, and data streams through blockchain-verified updates
   - Selects optimal routes by analyzing historical patterns and current network conditions

2. Resource State Synchronization
   - Maintains unified records across three blockchain types (material, financial, informational)
   - Updates all systems simultaneously through time-sensitive verification protocols

The Logisticon serves as the semantic translation engine that:

1. Digital Process Modeling
   - Converts physical warehouse operations into AI-readable formats
   - Creates interlinked digital models of supplier networks and distribution channels

2. Scenario Prediction

- Forecasts potential disruptions using supply chain simulations
- Generates multiple "what-if" scenarios for risk mitigation

3. Self-Regulating Allocation

- Automates inventory distribution through smart contracts
- Maintains optimal stock levels using live consumption data

The key Synchronization Features and Security Implementation aspects related to the integrated Cognitome-Logisticon framework:

Synchronization Features

1. Real-Time Twin Coordination

- The Cognitome ensures continuous alignment between physical operations and their digital counterparts using time-sensitive blockchain updates
- The Logisticon classifies these updates into standardized categories for AI processing

2. Emergency Response Mechanism

- During disruptions, the Cognitome activates temporary blockchain networks for urgent data processing
- The Logisticon translates these emergency protocols into operational guides for human teams

3. Multi-Level Governance

- The Cognitome enforces policy hierarchies from local to global scales through blockchain consensus
- The Logisticon maintains clear audit trails of governance decisions across all levels

Security Implementation

1. Resource-Specific Protection

- The Cognitome operates distinct blockchain systems for tracking goods (material), payments (financial), and contracts (informational)
- The Logisticon categorizes security requirements for each resource type in AI-friendly formats

2. Mandatory Cross-Verification

- Every transaction must connect to all three blockchain types simultaneously (goods+payments+contracts)
- The Logisticon documents these interconnections in machine-readable formats for compliance checks

3. Adaptive Data Protocols

- The Cognitome switches between high-detail and low-detail sharing modes based on network conditions
- The Logisticon automatically adjusts security clearance levels during these transitions

This integrated approach allows:

- Continuous status updates across physical and digital systems through coordinated blockchain updates
- Crisis-proof operations via emergency blockchain networks that bypass normal protocols
- Seamless policy enforcement across different organizational scales
- Tamper-proof records through mandatory triple-ledger verification for all transactions
- Context-sensitive security that tightens during emergencies and relaxes during routine operations

The Cognitome handles the technical execution of these features through blockchain management and predictive algorithms, while the Logisticon ensures operational transparency by translating technical processes into standardized guidelines and audit records

Together, these layers form a unified hypernetwork that ensures secure, real-time data exchange and adaptive, intelligent coordination.

## 6    Hierarchical Cognitum Hypernetwork for Smart City Ecosystems and Beyond

Our framework extends to hierarchical systems that include:

1. **Individual Agents:** Humans and AI entities generating data and executing decisions.
2. **Corporate Entities:** Aggregates of individuals coordinating tasks and sharing resources.
3. **Cities:** Self-regulating clusters serving as local nodes.
4. **Countries:** National-level entities enforcing standardized policies and aggregating data.
5. **Global Civilization:** An overarching decentralized cognitive network integrating diverse national systems.

Each level employs a standardized multi-blockchain ecosystem to maintain immutable records, supporting decentralized governance and global coordination.

## 7    AI-Driven Digital Twins and Real-Time Data Integration

Building on the theoretical framework in Sections 2–6, this section illustrates the application of our approach to digital twin technology for urban simulation. It outlines how AI-driven digital twins are enhanced by our hypernetwork and blockchain architecture, ensuring real-time data synchronization and predictive analytics.

### 7.1    Digital Twin Architecture

Digital twins serve as virtual replicas of urban systems, capturing the dynamics of physical processes. Our framework integrates these digital twins by:

- **Modeling Urban Environments:** Digital twins simulate infrastructure, transportation, and resource flows with high fidelity.
- **Temporal Consistency:** The time-exact blockchain synchronization ensures that the state of the digital twin is updated in real time, reflecting changes from the physical world.
- **Error Bounds and Confidence Metrics:** AI prediction models within the digital twin architecture quantify uncertainties, providing confidence metrics that help optimize decision-making.

### 7.2    AI Prediction Models

Our framework leverages advanced AI models to forecast urban dynamics:

- **Model Architecture:** Deep learning models process high-fidelity data from IoT sensors and simulation engines to predict traffic patterns, energy usage, and resource demands.
- **Error Bounds and Confidence Metrics:** These models incorporate statistical error bounds and confidence intervals, ensuring reliable predictions that inform real-time urban management.
- **Real-Time Update Scheme:** Data pipelines integrated with time-exact blockchain synchronization ensure that predictions are updated continuously, allowing for adaptive planning.

### 7.3 REAL-TIME DATA INTEGRATION

Seamless data integration is crucial for digital twins:

- **Data Ingestion Pipeline:** Sensor data, transactional records, and simulation outputs are aggregated and processed in real time.

- **State Synchronization:** The hypernetwork ensures that digital twin states remain synchronized with the physical world through immutable blockchain records.

- **Adaptive Feedback Loops:** AI-driven analysis provides feedback to adjust urban models, enhancing the digital twin's accuracy and reliability.

## 8 SYSTEM DESIGN AND IMPLEMENTATION

This section outlines the overall system architecture, development environment, and prototype configuration that bring our theoretical framework into practice. It builds on the theoretical and application layers described in Sections 2–7.

### 8.1 SYSTEM ARCHITECTURE

The system is composed of several integrated components:

- **Hypernetwork Backbone:** Integrates material, financial, and informational flows via specialized blockchain ecosystems.

- **Digital Twin Module:** Simulates urban environments using AI-driven models, synchronized in real time.

- **Cognitive and Semantic Layers:** The cognitum and logisticon provide adaptive control and regulatory oversight.

- **Data Flow Pipeline:** IoT sensors and simulation engines feed data into the hypernetwork, ensuring continuous state updates.

### 8.2 DEVELOPMENT ENVIRONMENT AND TOOLS

Our prototype is developed using:

- **Blockchain Platforms:** Customized implementations for global, regional, and local layers.

- **AI Frameworks:** TensorFlow and PyTorch for developing prediction models.

- **Simulation Tools:** Urban simulation software integrated with digital twin frameworks.

- **Integration Middleware:** APIs and real-time data processing frameworks for IoT and sensor data.

### 8.3 PROTOTYPE CONFIGURATION

Key aspects of our prototype include:

- **Modular Design:** Each component (blockchain, AI, simulation) is developed as a modular unit to facilitate testing and scalability.

- **Interoperability:** Standardized data formats and protocols ensure seamless integration across different layers and components.

- **Security and Privacy:** Robust encryption and authentication mechanisms protect data integrity and ensure compliance with privacy standards.

## 8.4 Evaluation Metrics and Scenarios

The evaluation system was built on theoretical analysis and scenario-based assessment to establish performance indicator for model optimizations. A set of performance vectors and evaluation criteria are introduced to assess the system's feasibility, potential performance, and theoretical bounds.

A performance quadrant system categorizes performance by utilizing *temporal efficiency* dimension and *system adaptability* dimension. Temporal efficiency distinguishes between fast-impact vectors for system behavior and vectors that accumulate influence over longer time horizon. Vectors on the left side of the quadrant have short-term impact in real-time operations. Vectors on the right side have long-term impact, which form evolutionary from cumulated learning, adaptation and scaling. System adaptability contains dynamic vectors, divided to highly adaptable and responsive vectors, and pseudo-static vectors with low adaptability and high infrastructure-dependendancies. High-adaptability vectors involve constantly evolving and operational vectors while vectors with low-adaptability are bound to scalability and security iterations. Fig. 1 illustrates the elements of performance vector quadrant.

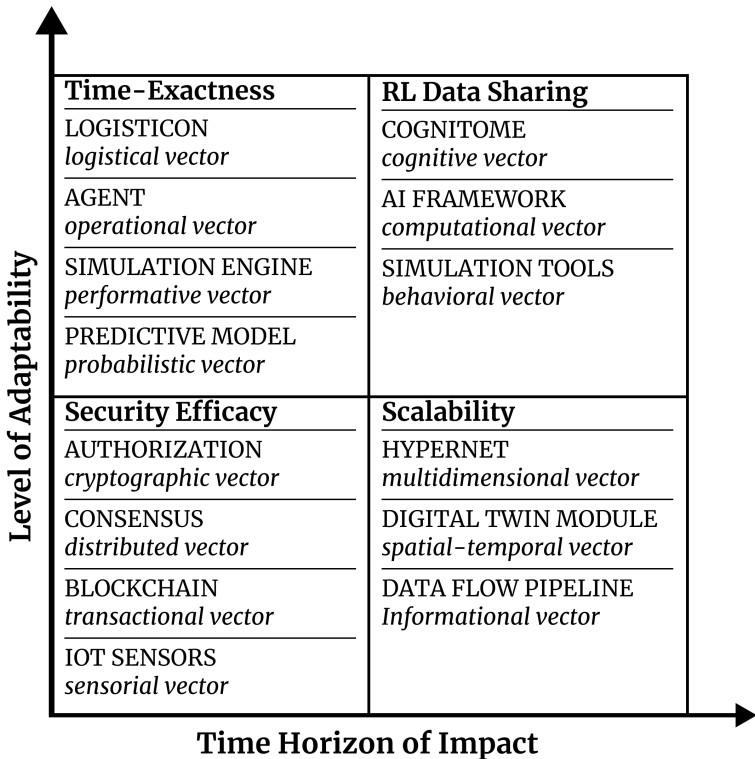

Figure 1: Performance Vectors Quadrant components and associated vectors.

Overall performance for is calculated by:

$$Q_{\text{total}} = Q_I + Q_{II} + Q_{III} + Q_{IV},$$

where quadrant values are formed from combinations of $T_{\text{short}}$ (short-term temporal vector), $T_{\text{long}}$ (long-term temporal vector), $A_{\text{low}}$ (low-adaptability vector), and $A_{\text{high}}$ (high-adaptability vector). Each quadrant is aggeregated as follows:

$$Q_I = \sum_i (T_{\text{short}} \times A_{\text{high}})_i \qquad \text{Short-term impact, High adaptability (Quadrant I)}$$

$$Q_{II} = \sum_j (T_{\text{short}} \times A_{\text{low}})_j \qquad \text{Short-term impact, Low adaptability (Quadrant II)}$$

$$Q_{III} = \sum_k (T_{\text{long}} \times A_{\text{high}})_k \qquad \text{Long-term impact, High adaptability (Quadrant III)}$$

$$Q_{IV} = \sum_l (T_{\text{long}} \times A_{\text{low}})_l \qquad \text{Long-term impact, Low adaptability (Quadrant IV)}$$

In summary, the proposed evaluation framework, centered on a performance quadrant system, provides a structured and comprehensive approach to evaluating model optimizations. By dissecting performance into temporal efficiency and system adaptability, and quantifying these dimensions through specific vector combinations, analysis of both immediate and long-term impacts can be conducted systematically. This methodology allows for understanding and monitoring of system behavior, enabling targeted improvements and ensuring that models are not only efficient in the short term but also robust and adaptable over extended periods.

# 9 MATHEMATICAL FORMULATION AND ALGORITHMIC SYNTHESIS

For each ecosystem $x \in \{m, f, i\}$, let $B_{x,\alpha}(t)$ denote the state of blockchain $\alpha$ at time $t$. Its evolution is given by:

$$h_{x,\alpha}^{(t+1)} = H\left(h_{x,\alpha}^{(t)}, \phi_{x,\alpha}(t), \Psi_x(t)\right),$$

**Polynomial-Time Verification:** To ensure time-exact processing and scalability, each blockchain state update and verification is designed to operate in polynomial time Goncharov et al. (2024); Goncharov & Nechesov (2022). This means the verification algorithm runs in $O(n^k)$ time (for some fixed constant $k$), ensuring that even as the volume of data increases, the process remains computationally efficient.

## 9.1 ADAPTIVE DATA FIDELITY AND DYNAMIC BLOCKCHAIN ASSEMBLY

Each agent $i$ maintains a state:

$$s_i(t) = \{p_i(t), T_i(t), E_i(t), L_i(t), S_i(t)\},$$

where $p_i(t)$ is the agent's position, $T_i(t)$ is its current task, $E_i(t)$ is an event indicator, $L_i(t)$ denotes network load, and $S_i(t)$ is the reliability score. The RL policy selects an action $a_i(t)$ that determines the data fidelity mode:

$$f_i(t) = \begin{cases} 0, & \text{if low-fidelity (textual) sharing is selected,} \\ 1, & \text{if high-fidelity (rich data) sharing is selected.} \end{cases}$$

The communication cost is modeled as:

$$C_i(t) = f_i(t)\, C_{\text{high}} + \left(1 - f_i(t)\right) C_{\text{low}}.$$

When $f_i(t) = 1$, the routing function:

$$\mathcal{R}(E, p, r) = \{j \in A : d(p_j, p_E) \leq \delta \,\wedge\, R_j \geq r_{\min}\}$$

selects agents to form a temporary blockchain $B_{\text{temp}}$ for processing high-fidelity data.

## 9.2 HIERARCHICAL AGGREGATION AND GLOBAL CONSENSUS

For hierarchical layers $k$ (e.g., individuals, corporations, cities, countries, global civilization), the aggregated state is:

$$A_k(t+1) = \text{Aggregate}\left(\{h_{x,\alpha}^{(t+1)} : \text{agents in layer } k\}\right).$$

where

$$\text{Aggregate}(\{h_i\}) = \frac{1}{N} \sum_{i=1}^{N} h_i$$

The global state is computed as either a weighted sum:

$$G(t+1) = \sum_{k=1}^{K} w_k \, A_k(t+1),$$

## 9.3 DECENTRALIZED COGNITIVE AND SEMANTIC COORDINATION

A decentralized cognitive layer computes:

$$C(t+1) = \mathcal{C}\Big(H(t+1)\Big)$$

for some fixed cognitive algorithm $\mathcal{C}(x)$ which analyse conditions of the hypergraph $H$.
The semantic (Logisticon) layer enforces regulation:

$$S(t+1) = \mathcal{S}\Big(C(t+1), \, H(t+1)\Big).$$

for some fixed semantic algorithm $\mathcal{S}(x, y)$ that depends from cognitive parameter and hypergraph conditions.

## 9.4 Overall System Algorithm

---

**Algorithm 1** Algorithm 1: Holistic Global Cognitome Hypernetwork System

---

1: **for** each time step $t \geq 0$ **do**
2:     **for** each ecosystem $x \in \{m, f, i\}$ and each blockchain $B_{x,\alpha}$ **do**
3:         Update blockchain state:

$$h_{x,\alpha}^{(t+1)} = H\left(h_{x,\alpha}^{(t)}, \phi_{x,\alpha}(t), \Psi_x(t)\right)$$

4:     **end for**
5:     Form node sets:

$$V_x(t+1) = \bigcup_{\alpha \in J_x} V_{x,\alpha}(t+1)$$

6:     Construct hypergraph:

$$H(t+1) = \left(\bigcup_x V_x(t+1),\ E(t+1)\right),$$

where

$$E(t+1) = \{e \subset \bigcup_x V_x(t+1) : e \cap V_m(t+1) \neq \varnothing,\ e \cap V_f(t+1) \neq \varnothing,\ e \cap V_i(t+1) \neq \varnothing\}$$

7:     **for** each agent $i$ **do**
8:         Observe state:

$$s_i(t) = \{p_i(t), T_i(t), E_i(t), L_i(t), S_i(t)\}$$

9:         Select action $a_i(t)$ via RL and update $Q(s, a)$.
10:        Determine fidelity $f_i(t)$ and compute cost:

$$C_i(t) = f_i(t)\, C_{\text{high}} + (1 - f_i(t))\, C_{\text{low}}$$

11:        **if** $f_i(t) = 1$ **then**
12:           Execute routing:

$$\mathcal{R}(E, p, r) = \{j \in A : d(p_j, p_E) \leq \delta \ \wedge\ R_j \geq r_{\min}\}$$

13:           Form temporary blockchain $B_{\text{temp}}$ among selected agents.
14:        **end if**
15:     **end for**
16:     **for** each hierarchical layer $k = 1, \ldots, K$ **do**
17:         Compute aggregated state:

$$A_k(t+1) = \text{Aggregate}\left(\{h_{x,\alpha}^{(t+1)} : \text{agents in layer } k\}\right)$$

18:     **end for**
19:     Derive global state:

$$G(t+1) = \sum_{k=1}^{K} w_k\, A_k(t+1)$$

20:     Compute cognitive state:

$$C(t+1) = \mathcal{C}\left(H(t+1)\right)$$

21:     Derive semantic regulation state:

$$S(t+1) = \mathcal{S}\left(C(t+1), H(t+1)\right)$$

22:     Update all states and proceed to $t + 1$.
23: **end for**

---

## 10  CONCLUSION

We have presented a comprehensive framework for the logistics and service metaverse that integrates hypernetworks with multi-blockchain architectures and decentralized cognitive control. By unifying material, financial, and informational flows into a cohesive hypergraph and incorporating adaptive data fidelity and dynamic stakeholder routing, our approach lays a robust foundation for secure, real-time digital twin environments. The cognitome, with its predictive management capabilities, enables proactive decision-making by forecasting process developments and optimizing resource allocation. In parallel, the proposed concept of logisticon serves as a semantic framework that translates complex logistic operations into structured language models, thereby bridging physical operations with advanced cognitive technologies. The integration of the cognitome and the logisticon provides both a cognitive and semantic framework essential for next-generation smart city innovations and beyond.

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
