# OpenReview forum: "Metaverse Architectures: Hypernetwork and Blockchain Synergy"
_mathai.club/MathAI/2025/Conference — MathAI 2025 Oral_

### Official Review · Reviewer_x6Qe · 2025-02-26
**METAVERSE ARCHITECTURES: HYPERNETWORK AND BLOCKCHAIN SYNERGY. Reviewer recommends to include it in the Program of the International conference “Mathematics of Artificial Intelligence” (24-28 March 2025, Sochi) with its publication.**

**Rating:** 9
**Confidence:** 4

**Review:**

Modern urban environments are increasingly driven by digital transformation, where physical and virtual processes must be managed seamlessly. To integrate such complex processes, novel frameworks are needed that unify disparate data flows and management systems. The anonymous authors of this article propose a Hypernetwork-driven digital twin framework for the metaverse, integrating AI-driven predictions with time-exact blockchain synchronization to enhance urban modeling accuracy and virtual ecosystem performance. The proposed system employs a hierarchical blockchain architecture with temporal verification mechanisms to ensure secure and scalable data exchange in real time. The authors tackle critical challenges in resource interaction, security, and privacy within metaverse-based infrastructures by embedding polynomial-time verification into IoT data pipelines and simulation engines. The proposed approach establishes a trustworthy, adaptive, and decentralized ecosystem for digital twins, advancing next-generation smart city innovations.
        It should be specially noted that the proposed framework and the article in the whole are well structured to facilitate the acquaintance of readers with this multifaceted work. To understand the described approach, the authors propose the definitions of Metaverse, Hypernetworks, Cognitome and a new concept of Logisticon. Traditional economic research often treats material, financial, and informational flows in isolation, which impedes the development of comprehensive management models. The article addresses this gap by proposing a unified framework that merges these flows, enabling advanced AI integration for predictive and adaptive system management. By unifying material, financial, and informational flows into a cohesive hypergraph and incorporating adaptive data fidelity and dynamic stakeholder routing, the proposed approach lays a robust foundation for secure, real-time digital twin environments. The proposed concept of logisticon serves as a semantic framework that translates complex logistic operations into structured language models, thereby bridging physical operations with advanced cognitive technologies. The integration of the cognitome and the logisticon provides both a cognitive and semantic framework essential for next-generation smart city innovations and beyond.
       The list of References includes 17 appropriate literary sources to allow becoming familiar with many additional details of the approach presented. The quality, clarity, originality and significance of this work are high.
       I like this article and I recommend to include it in the Program of the International conference “Mathematics of Artificial Intelligence” (24-28 March 2025, Sochi) with its publication.

---

### Official Review · Reviewer_am1S · 2025-02-27
**Rewiew for paper 9**

**Rating:** 7
**Confidence:** 3

**Review:**

This paper presents the integration of hypernetworks with multi-blockchain architectures and decentralized cognitive control. Holistic Global Cognitome Hypernetwork System is proposed by the authors. A hypergraph is introduced to describe the integrated ecosystem.

The paper is easy to read and has clear sections, but the following points still need to be improved:
- References are introduced unlikely so it is not always clear what some reference refers to.
- For a uniform structure, I would highlight the concept of Cognitome on line 91. Overall, the structure and layout of the article can be considered an asset.
- Typo on line 222.
- Repetition on line 490.
- The section on System design mentions AI-driven models but no specific model.
- Metrics in 8.4 are not formalizable.

Possible recommendation: In section 9, the formulas could be numbered so as not to repeat them when there is a need to return to them.

This is a good structured and theoretically prepared paper, but it lacks practical results and metrics.

---

### Official Review · Reviewer_dHmN · 2025-02-27
**Metaverse Architectures: Hypernetwork and Blockchain Synergy**

**Rating:** 9
**Confidence:** 4

**Review:**

The article proposes a digital twin framework based on hypernetworks and blockchain, demonstrating strong innovation. It integrates material, financial, and informational flows using hypergraphs, leveraging mechanisms such as reinforcement learning and adaptive data fidelity to ensure the system can adjust flexibly. The framework also establishes cognitive and logisticon layers, enabling the conversion from operations to language models and facilitating proactive decision-making. The proposed framework can enhance the accuracy of urban modeling and the performance of virtual systems. The article is well-structured, covering various aspects such as theoretical framework, system design, and algorithm implementation, with each section providing a clear and organized introduction. The system architecture is rationally designed and has practical application potential.
If experimental simulations are added to validate the effectiveness of the proposed methods based on the existing theoretical framework, it could further enhance the credibility of the research.

---

### Decision · Program_Chairs · 2025-03-08

**Decision:**

Accept (Oral)

**Comment:**

Your article has been accepted and you can give a talk on the article. All articles will be sorted by rating and within the available conference places one author from each article will be invited. If there are not enough places, then you will either have the opportunity to speak remotely or come at your own expense!